# Flame Anchoring of an $H_2/O_2$ Non-Premixed Flame with $O_2$ Transcritical Injection

**Eugenio Giacomazzi** *,†,‡ , **Donato Cecere** ‡ **and Nunzio Arcidiacono**

Laboratory of Processes & Systems Engineering for Energy Decarbonisation, ENEA, 00123 Rome, Italy
* Correspondence: eugenio.giacomazzi@enea.it
† Current address: Casaccia Research Center, TERIN-PSU-IPSE, S.P. 081, ENEA, Via Anguillarese 301, S.M. Galeria, 00123 Rome, Italy.
‡ These authors contributed equally to this work.

**Abstract:** The article is devoted to the analysis of the flame anchoring mechanism in the test case MASCOTTE C-60 RCM2 on supercritical hydrogen/oxygen combustion at 60 bar, with transcritical (liquid) injection of oxygen. The case is simulated by means of the in-house parallel code HeaRT in the three-dimensional LES framework. The cubic Peng–Robinson equation of state in its improved translated volume formulation is assumed. Diffusive mechanisms and transport properties are accurately modeled. A finite-rate detailed scheme involving the main radicals, already validated for high-pressure $H_2/O_2$ combustion, is adopted. The flow is analysed in terms of temperature, hydrogen and oxygen instantaneous spatial distributions, evidencing the effects of the vortex shedding from the edges of the hydrogen injector and of the separation of the oxygen stream in the divergent section of its tapered injector on the flame anchoring and topology. Combustion conditions are characterised by looking at the equivalence ratio and compressibility factor distributions.

**Keywords:** transcritical flows; high pressure; non-premixed flames; oxy-combustion; hydrogen





## 1. Introduction

Aerospace applications, and in particular liquid oxygen rocket engines and cooling systems, have largely promoted research on real gas flows. In recent decade, more attention has been focused on high-pressure combustion of reactants exhibiting real gas behaviour with regard not only to the renewed interest in space exploration, but also its application in electric power generation, as in advanced supercritical $CO_2$ gas turbine cycles (designed to operate at 300 bar) [1], organic Rankine cycles and diesel engines. In such applications, the flow can be far away from ideal thermodynamics and the different fluid behaviour has to be accounted for by means of real gas equations of state and specific models for molecular transport properties [2].

Experimental work at such high-pressure conditions may be prohibitive: the use of advanced laser diagnostics is not an easy task to achieve (few examples exist); in addition, facilities themselves may be very expensive. The literature reports experiments on transcritical/supercritical single and coaxial jet configurations. Non-reacting studies focus on the estimation of the dense core length of the central jet and its growth rate (or spreading angle) [3,4], its fractal dimension and length scales [5], and on the interaction with external acoustic fields [6]. Reacting conditions are mainly studied by means of shadowgraphy and imaging of the flame emission (LIF technique focusing on OH or CH radicals) for qualitative characterisation of the near injector process [7,8]; very few experiments provide velocity and/or temperature data [9,10].

Most research and design in transcritical/supercritical combustion is based on numerical simulation, but also this approach is not an easy task.

Some issues deal with the modeling accuracy of the equation of state, e.g., its ability to capture huge variations of fluid properties when crossing the pseudo-boiling line; some

others deal with its computational efficiency; i.e., too complex and accurate equations of state (EoSs) cannot be used in time-consuming approaches, such as LES and DNS. In fact, some EoSs can be very accurate at the expense of a non-straightforward solution of the fluid state, like the GERG2004 [11], the Benedict–Webb–Rubin EoS or its modified version [2] (Chp. 4.7). For LES or DNS, approaches for which an analytical solution is readily available (ideal gas, real gas cubic EoS, such as Van der Waals, Redlich–Kwong and its Soave modification, Peng–Robinson, etc.) [2] (Chp. 4.6) would be preferable; this justifies the adoption of a cubic EoS for this work.

In addition, critical uncertainties arise from the chemical kinetics mechanisms: researchers compare different detailed mechanisms, try to identify the best one by means of the few high-pressure experimental data and derive reduced ones for fluid dynamic simulations [12,13].

Furthermore, numerical schemes are stressed: the high-density gradients typical of some applications (as those involving liquid injection) and the multi-species transport enhance wiggle formation in fully compressible solvers; spurious pressure oscillations are generated when a fully conservative scheme is adopted, due to the high non-linearity of the real-fluid equation of state. Different approaches are adopted [14]. Most authors use high-order low-dissipation centered schemes coupled with specifically designed artificial viscosity terms consistently applied to all transported variables to avoid the growing of strong unphysical acoustic waves [15]; in addition, the total energy transport equation can be replaced by a pressure equation, as in [16]. WENO formulations can also be adapted to real gases in approximate Riemann solvers [17], transporting an additional equation for the specific heat ratio (starting from a double-flux approach) to prevent pressure errors from material and contact discontinuities [18].

Due to the complexity of the problem, most of the articles in the literature are aimed at comparing numerical predictions with experimental data for the available test cases. Concerning high-pressure real gas oxy-combustion, some authors analysed subgrid scale contributions [19,20] or compared different combustion models [21]; some others investigated features of high-pressure flames [13,22,23], evidencing how much has still to be done.

This work is devoted to the numerical simulation of the $H_2/O_2$ combustion in the MASCOTTE facility with transcritical injection of $O_2$ and in particular the analysis of its flame anchoring mechanism. With respect to past works, the effect of heat transfer into the solid body of the injector is included to have more accurate wall boundary conditions. Flame and shear-layer dynamics is described and the structure of the liquid oxygen jet and of the flame analysed to identify the regions where real gas compressibility has an important role.

## 2. The Numerical Code: Heart

In this work the compressible Navier–Stokes equations are solved for a reacting real gas flow in the Large Eddy Simulation framework. The mathematical models adopted are derived for a Newtonian and Stokesian fluid of $N_s$ chemical species.

### 2.1. The Real Gas Equation of State

The Peng–Robinson cubic equation of state (EoS) in its improved translated volume formulation is assumed [24,25]:

$$p = \frac{R_u T}{v + c - b_m} - \frac{a_m}{(v + c)(v + c + b_m) + b_m(v + c - b_m)} , \tag{1}$$

where $R_u$ is the universal gas constant, $v$ the molar volume and the $a_m$ [J m$^3$ mol$^{-2}$] and $b_m$ [m$^3$ mol$^{-1}$] coefficients are functions of composition and temperature, and $c$ is the translation parameter that effectively shifts or translates each $v$ term in the EoS. The real gas EoS differs from the ideal law because it takes into account some inter-molecular forces: the first term (instead of the ideal $R_u T/v$) models the repulsive force that molecules exert on each other at short distance and the coefficient $b_m$ is proportional to the actual

volume of the molecule; the second additional term models the long-range attractive forces between the molecules such as electrostatic forces, polarisation or London dispersion forces, i.e., the forces that keep a gas together and that allow the liquid state to exist (they decrease the pressure exercised by the fluid on the walls of a vessel, hence the negative sign for this term).

For modeling $a_m$ and $b_m$, the Peng–Robinson EoS requires only the acentric factor on top of the critical properties and it is thus easy to implement for a wide range of species. Critical data for the species involved in the present simulations are reported in Table 1.

**Table 1.** Critical properties of the species considered in the present article; $T_c$ $P_c$, $V_c$ and $\omega$ are, respectively, the critical temperature, pressure, molar volume and acentric factor. Sources: [26] for $N_2$, $H_2$, $O_2$ and $H_2O$; [27] for H, O, OH and $H_2O_2$. $HO_2$ data were estimated through the CRANIUM code [28] that is based on the molecular description and group contribution strategies: Joback's method [29] for critical temperature and volume and Lydersen's method [30] for critical pressure; Pitzer's expression of vapor pressure for the acentric factor [2] (Sections 2.3, 7.2 and 7.4).

| Species | $T_c$ (K) | $P_c$ (Bar) | $V_c$ (m$^3$/mol) | $\omega$ (−) |
|---|---|---|---|---|
| $N_2$ | 126.19 | 33.958 | $8.941 \times 10^{-5}$ | 0.0372 |
| $H_2$ | 33.190 | 13.150 | $6.693 \times 10^{-5}$ | −0.2140 |
| $O_2$ | 154.58 | 50.430 | $7.337 \times 10^{-5}$ | 0.0222 |
| $H_2O$ | 647.09 | 220.64 | $5.595 \times 10^{-5}$ | 0.3443 |
| OH | 100.70 | 57.600 | $3.930 \times 10^{-5}$ | 0.3290 |
| O | 100.70 | 57.600 | $3.930 \times 10^{-5}$ | 0.3290 |
| H | 182.60 | 25.190 | $1.630 \times 10^{-5}$ | 0.3290 |
| $HO_2$ | 472.40 | 106.65 | $6.350 \times 10^{-5}$ | 0.6873 |
| $H_2O_2$ | 587.19 | 93.530 | $7.350 \times 10^{-5}$ | 1.0200 |

For a mixture including $N_s$ species the coefficient $b_m$ is calculated as

$$b_m = \sum_{i=1}^{N_s} x_i b_i \, , \tag{2}$$

where $x_i$ is the mole fraction of species i. The individual species coefficient $b_i$ is given by

$$b_i = 0.0077796 \frac{R_u T_{c_i}}{p_{c_i}} \, , \tag{3}$$

where $p_{ci}$ and $T_{ci}$ are the $i$-th species critical pressure and temperature.

Among the cubic EoS, the Peng–Robinson as well as the Redlich–Kwong EoS and its Soave modification, model the cohesive energy parameter $a_m$ from the critical properties: $a_m = \alpha a_m(T_c, p_c)$. However, while for the RK-EoS $\alpha = 1$, for both the PR and SRK-EoS $\alpha$ is a temperature-dependent function involving the acentric factor too, which is a measure of the non-sphericity of the molecules. As a consequence, energies and specific heat predictions are more accurate, an important feature for reacting flows. For a mixture including $N_s$ species, the coefficient $a_m$ is calculated as:

$$a_m = \sum_{i=1}^{N_s} \sum_{j=1}^{N_s} x_i x_j a_{ij} \, . \tag{4}$$

The binary coefficients $a_{ij}$ are estimated as

$$a_{ij} = (1 - k_{ij})(a_i a_j)^{1/2} \, , \tag{5}$$

$k_{ij}$ being the binary interaction parameters (taking into account non-linear intermolecular effects) tabulated [31] or empirically calculated [32], i.e.,

$$1 - k_{ij} = \left( \frac{2V_{c_i}^{1/6} V_{c_j}^{1/6}}{V_{c_i}^{1/3} + V_{c_j}^{1/3}} \right)^3 , \tag{6}$$

where $V_c$ is the individual species critical molar volume. The individual species coefficients $a_i$ are given by

$$a_i = 0.457236 \frac{(R_u T_{c_i})^2}{p_{c_i}} \alpha(T_{r_i}) , \tag{7}$$

with

$$\alpha(T_{r_i}) = 1 + f(\omega_i)\left(1 - T_{r_i}^{1/2}\right) , \tag{8}$$

where $T_{r_i} = T/T_{c_i}$ is the individual species reduced temperature and the function $f(\omega_i)$ is defined as

$$f(\omega_i) = \begin{cases} 0.374640 + 1.54226\omega_i - 0.26992\omega_i^2 & \text{for } \omega_i \leq 0.49 \\ 0.379642 + 1.48503\omega_i - 0.164423\omega_i^2 + 0.016666\omega_i^3 & \text{for } \omega_i > 0.49 \end{cases} \tag{9}$$

where $\omega_i$ is the $i$-th species acentric factor.

### 2.2. The Diffusive Fluxes

The mathematical models adopted are derived for a fluid of $N_s$ chemical species. The constitutive laws assumed to describe the behaviour of the fluid are here reported. They simply model the microscopic molecular diffusion of momentum, energy and mass, i.e., they model the momentum flux $S$, the heat flux $Q$ and the species mass flux $J_i$.

A Newtonian fluid is considered and the Stokes' assumption is made: it is characterised by the following constitutive relation between the stress, $S$, and the strain rate, $E$,

$$S = -(p + 2/3\,\mu\nabla \cdot \boldsymbol{u})\boldsymbol{I} + 2\mu E = -p\boldsymbol{I} + \mathcal{T} , \tag{10}$$

$\mu$ being the kinematic viscosity; $\mathcal{T}$ is the viscous part of the stress tensor.

The mass diffusion flux has three contributions [33]. The first one is due to concentration gradients (here modeled through the Hirschfelder and Curtiss' law for multicomponent mixtures) [34], the second due to pressure gradients (the baro-diffusion mechanism), and the third one due to temperature gradients (the thermo-diffusion or Soret effect) [35]:

$$\begin{aligned} \boldsymbol{J}_i = \rho Y_i \boldsymbol{V}_i &= \boldsymbol{J}_i^{HC} + \boldsymbol{J}_i^{BD} + \boldsymbol{J}_i^{S} \\ &= -\rho Y_i D_i \left[ \frac{\nabla X_i}{X_i} + \frac{X_i - Y_i}{X_i}\frac{\nabla p}{p} \right] - \mathcal{D}_i^T \frac{\nabla T}{T} . \end{aligned} \tag{11}$$

The diffusion coefficient $D_i$ is an *effective* diffusion coefficient of the $i$-th species into the mixture (mixture-average assumption). The thermo-diffusion, or Soret effect, is the mass diffusion due to temperature gradients, driving light species towards hot regions of the flow [36]. This effect, often neglected, is neverthless known to be important, in particular for hydrogen combustion and in general when very light species play an important role [37].

Keeping apart the radiative heat transfer of energy, the heat flux has three contributions too. The first is due to temperature gradients (the Fourier diffusion), the second is due to mass diffusion fluxes and the third one is the Dufour effect (reciprocal of the Soret effect):

$$Q = \boldsymbol{q}_F + \boldsymbol{q}_{V_i} + \boldsymbol{q}_D = -K\nabla T + \rho \sum_{i=1}^{N_s} h_{s_i} Y_i \boldsymbol{V}_i + \boldsymbol{q}_D. \tag{12}$$

Note that this is the heat flux expression entering into the energy transport equation where formation energies are isolated in a source term, i.e., not included in the energy definition. Usually the Dufour effect (the third term) is negligible even when thermo-diffusion is not [35] (p. 768) and hence it is neglected in the present work.

Molecular viscosity and thermal conductivity are accurately modeled through NIST models in REFPROP with an Extended Corresponding States method and fluid-specific correlations [38]. Since REFPROP considers only stable species, in this simulation only $H_2$, $O_2$ and $H_2O$ are assumed to contribute to the mixture viscosity and thermal conductivity; the other species, having much smaller mass and volume fractions, are assumed to have a negligible contribution. It is also stressed that for such unstable species, no reliable dipole moment data exist: these quantities are needed in common viscosity and thermal conductivity as in [39].

The diffusion coefficient $D_i$ of the $i$-th species into the rest of mixture is modeled according to the Hirschfelder and Curtiss expression [34], where the required binary diffusion coefficient is calculated by means of kinetic theory with Takahashi's correction for high pressure [40]. As an example, Figure 1 shows the correction factors of some $H_2$ binary diffusivities as a function of the mixture reduced temperature evaluated from an instantaneous flowfield of the present MASCOTTE test case: at low reduced temperature, the binary diffusivity is nearly reduced by a factor of 10. The thermo-diffusion coefficient $\mathcal{D}_i^T$ is estimated by means of the EGLIB routines.

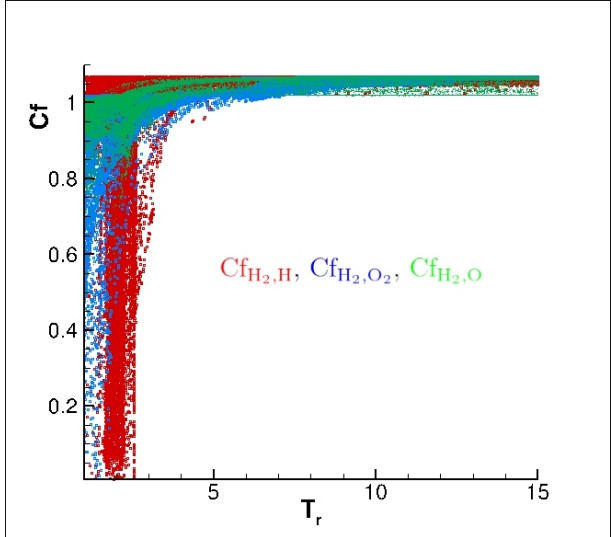

**Figure 1.** Takahashi's correction factor versus mixture reduced temperature for some hydrogen binary diffusion coefficients evaluated from instantaneous flowfield of the $H_2/O_2$ MASCOTTE combustion test case simulated in this article.

### 2.3. Turbulent Combustion Closure

In the non-reacting test case used for validation of the HeaRT code, the dynamic Smagorinsky subgrid scale model is adopted as turbulence closure. In the more complex reacting test case, unclosed turbulent combustion subgrid terms of the filtered compressible Navier–Stokes equations are modeled through Vreman's model [41] and the authors' LTSM (Localised Turbulent Scale Model) turbulent combustion model [42]. The eddy viscosity of Vreman's model is proportional to the gradient model of Clark [43] that outperforms most other models for supercritical mixing layers [44]. Compared to the Smagorinsky model, Vreman's approach has the advantage of vanishing in the laminar zones of the flow. In addition, its constant is not calculated dynamically: this is a great advantage since it avoids local instabilities due to eddy viscosity spotted patterns due to the dynamic procedure. In fact, the common implementation of the dynamic procedure incorporates explicit filtering operations, ensemble averaging in homogeneous directions and a somewhat ad hoc clip-

ping to prevent an unstable (negative) eddy viscosity. The extension of these techniques to complex flows is not trivial, which is an important reason to continue the search for an eddy viscosity that performs reasonably well without additional procedures.

According to LTSM, the Favre filtered chemical source term in the energy and single species transport equations is modeled as $\widetilde{\omega_i} \approx \gamma^* \omega_i^*$, $\gamma^*$ and $\omega_i^*$ being the local reacting volume fraction of the computational cell and the reaction rate of the $i - th$ chemical species. The subgrid reacting volume fraction is modeled as a premixed flamelet: given the local control volume $\mathcal{V}_\Delta$ with characteristic dimension $\ell_\Delta = \mathcal{V}_\Delta^{1/3}$, the LTSM assumes that it contains a flame front modeled with an actual thickness $\delta_F$, a laminar flame speed $\mathcal{S}_\mathcal{L}$ and a turbulent flame speed $\mathcal{S}_\mathcal{T}$, i.e.,

$$\gamma^* = \mathcal{G}_{ext} \frac{\mathcal{S}_\mathcal{T}}{\mathcal{S}_\mathcal{L}} \frac{\delta_\mathcal{F}}{\ell_\Delta} \,. \tag{13}$$

An extinction or flame stretch factor $\mathcal{G}_{ext} \leq 1$ is also included to take into account flame quenching due to subgrid scales. The subgrid flame front physics is synthesised in the flame speed ratio: the subgrid flame may be laminar or turbulent, wrinkled or not, thickened by turbulence or not, depending on the local filtered conditions of the flow: accordingly, LTSM models $\mathcal{S}_\mathcal{T}/\mathcal{S}_\mathcal{L}$ and $\delta_\mathcal{F}$.

*2.4. Numerical Schemes and Boundary Conditions*

The numerical simulations are performed by means of the in-house parallel code HeaRT and ENEA's supercomputing facility CRESCO [45].

The HeaRT code solves the compressible Navier–Stokes equations discretised through staggered finite difference schemes. A second-order accurate centered scheme is adopted for diffusive fluxes. Convective terms are discretised through the $AUSM^+$-$up$ method [46] coupled with a second-order accurate interpolation with a TVD, linear preserving limiter for non-uniform grids [47] to reduce spurious oscillations. It is noted that in simulations of transcritical liquid flows this methodology is not sufficient to avoid large spurious pressure oscillations along the liquid interface due to the compressible solver; according to the observations in [27], numerical dissipation is locally increased by forcing such a limiter to be first-order accurate when the fluid is in a compressible liquid state, i.e., the compressibility factor is less than 0.8. The low-storage third-order accurate Runge–Kutta method of Shu–Osher is used for time integration.

The total energy is defined as the sum of internal (thermal) and kinetic energy only. The authors found this choice mandatory [48,49] to avoid, or at least reduce, unphysical energy and temperature oscillations, mainly driving to the divergence of calculation. No spurious waves were experienced in previous simulations of premixed flames, when the total energy was defined including the chemical formation contribution. Heat and chemical source terms are treated implicitly in order to reduce equation stiffness [50].

Non-reflecting boundary conditions [51–53] are implemented at open boundaries in their extended form to take into account variable transport properties [54], local heat release [55] and real gas effects [56]. The assumed value for the relaxation constant in the partially non-reflecting treatment of the outlet is the theoretical value 0.27 [51,57]. Turbulent velocity fluctuations are superimposed to the mean inlet values, modeled by means of a synthetic turbulence generator [58].

## 3. Preliminary Validation of the HeaRT Code in Real Gas Simulations

Mayer's measurements [59] on nitrogen injection at real gas conditions into a cylindrical chamber initially filled in with nitrogen at ambient temperature are commonly adopted to validate numerical codes. The geometry and sizes of the cylindrical chamber and injector are shown in Figure 2.

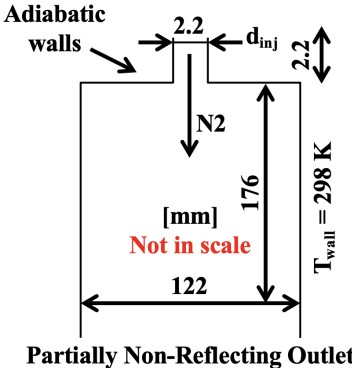

**Figure 2.** Sketch of the computational domain in the non-reacting Mayer's test case.

Here, Mayer's test case number 3 is chosen for the validation test since it is the most sensitive case in terms of thermophysical modeling: the $N_2$ jet is transcritical, thus exhibiting strong density spatial gradients and property changes, as shown in Figure 3. The chamber is initially filled in with gaseous $N_2$ at 298 K and 39.7 bar. Liquid $N_2$ is injected into the chamber at 126.9 K and $U_{inj} = 4.9$ m/s; the associated jet Reynolds number is 170,000. Since no measurements of the velocity fluctuation at the inlet of the chamber are provided, isotropic turbulence was artificially generated in the simulation imposing at the inlet a velocity fluctuation $u' = 2.5\% \, U_{inj}$ and a streamwise correlation length scale, $\ell_z = 0.132\,\mu m$. The fluctuation imposed is a typical value for turbulent round jets and was also adopted in [15].

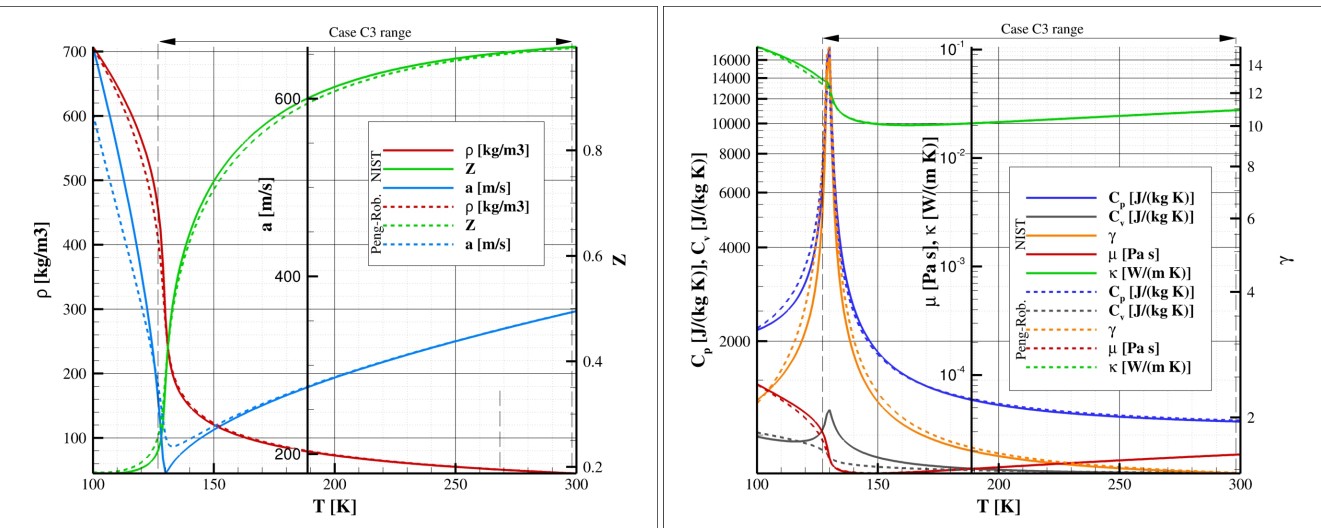

**Figure 3.** Comparison of some real gas properties estimated through the translated volume Peng–Robinson EoS and NIST reference data for $N_2$ at 39.7 bar as in the non-reacting Mayer's test case: density $\rho$, sound speed a, compressibility factor Z, specific heat ratio $\gamma = C_p/C_v$, viscosity $\mu$, thermal conductivity $\kappa$.

The three-dimensional cylindrical computational grid has $\sim$12.6 million nodes with 32 nodes along the azimuthal direction and consists of two zones: the first for the injection pipe has $18 \times 11$ nodes along the streamwise $z$ and radial $r$ direction; the second for the mixing chamber has $983 \times 400$ nodes. The covered sizes are reported in Figure 2. The grid is more refined close to the injection: $\Delta z \sim 0.13$ mm for $z < 26.67$ mm ($z = 0$ at the exit of the injector) and $\Delta r \sim 0.1$ mm for $r < 4.46$ mm.

The liquid nitrogen jet penetrates into the mixing chamber and the surge of $C_p$ across the pseudo-boiling line (see Figure 3, right) contributes to increasing the dense core penetration by increasing its resistance to heat transfer. Numerical simulation shows that toroidal

vortices are shed from the $N_2$ injector at 3.6 kHz (Strouhal number 0.22, defined through the dominant acoustic frequency, the average shear-layer thickness and the average maximum streamwise velocity close to injection); these coherent structures develop to turbulence at $\sim$20 mm from the injector after a spiral motion phase, as shown in Figure 4. Average data were computed by collecting 180 samples in 31.78 ms, corresponding to three convection times of the $N_2$ stream over 23.26 jet diameters $d_{inj}$ at $U_{inj}$. Numerical results were compared with the available experimental measurements: Figure 5 shows the average density distribution along the chamber axis and the average liquid nitrogen jet spreading angle, as defined by the experimentalists in [59].

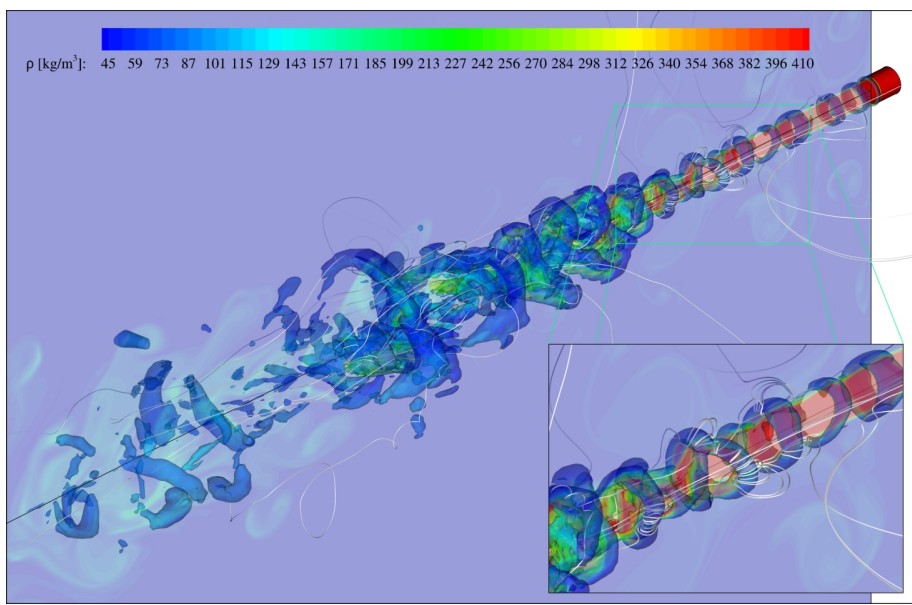

**Figure 4.** Instantaneous snapshot showing the spatial evolution of the coherent structures shed from the liquid $N_2$ injector in the Mayer test case. The vortex iso-surface identified by means of the Q-criterion are coloured by density levels.

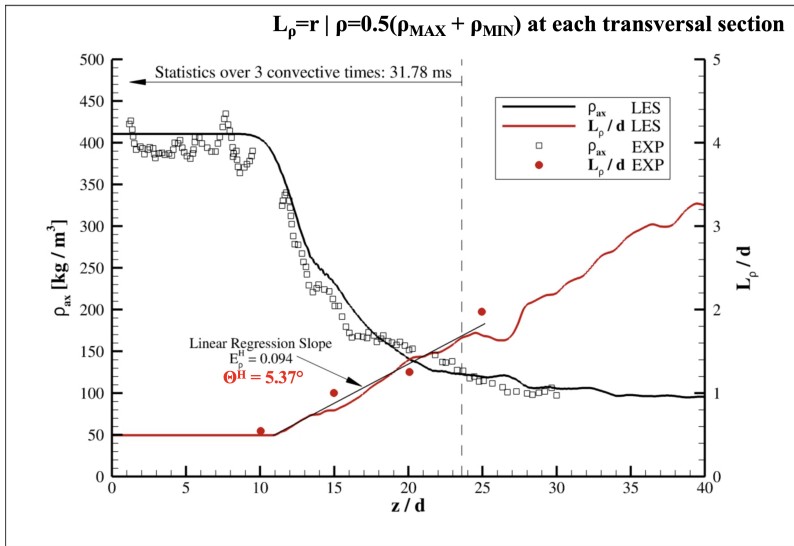

**Figure 5.** Comparison of numerical and experimental data of the Mayer test case 3: density distribution along the axis of the mixing chamber and liquid nitrogen jet spreading angle.

It is observed that the computational grid quality was checked by calculating the quality index defined by Pope [60], i.e., $IQ_{Pope} = \mathcal{K}_{turb}/(\mathcal{K}_{turb} + \mathcal{K}_{sgs})$. Such a check

results in $IQ_{Pope} > 84\%$ in most of the flow field, larger than the 80% limit suggested for a reliable LES grid.

## 4. Experimental Set-Up

The reactive case simulated in this work was experimentally investigated in the MAS-COTTE cryogenic combustion test facility developed by ONERA for rocket applications. It consists of a combustor having a 50 mm × 50 mm square-section, 40 cm length with a final nozzle and an axisymmetric coaxial injector for $O_2$ and $H_2$ or $CH_4$ mounted vertically downward, as sketched in Figure 6 (left).

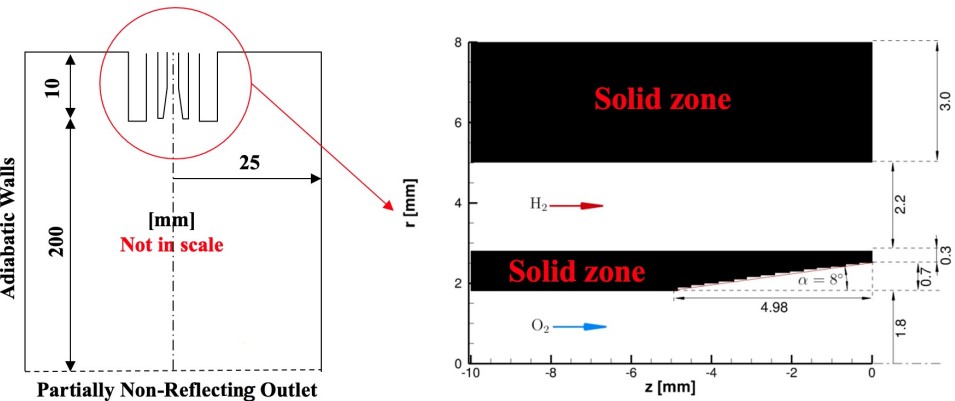

**Figure 6.** Sketch of the combustor and zoomed-in view of the tapered injector in the MASCOTTE C-60 test case, here modeled in a staircase fashion.

The particular test case numerically studied in this article is classified as RCM2: it has a nominal pressure of 60 bar (case C-60) and considers injection of transcritical (liquid) $O_2$ and gaseous $H_2$ [61,62]. The liquid oxygen is injected from the central pipe, which has a constant section of radius 1.8 mm before a linear divergent section with an angle of 8° and a length of 4.98 mm (the final radius is 2.5 mm); the end of the divergent is squared-off with a width of 0.3 mm. The hydrogen flows through the coaxial pipe having a radial width of 2.2 mm. The thickness of the wall separating the two jets is $h = 1$ mm. A zoomed-in-view sketch of the tapered injector is also shown in Figure 6 (right), as modeled in this work.

Flow injection data, including the turbulent velocity fluctuations forced at the inlets of the computational domain by means of a synthetic turbulence generator, are reported in Table 2. The reduced temperatures for $O_2$ and $H_2$ are 0.54 and 8.29, respectively; their reduced pressures are 1.19 and 4.56. The turbulent Reynolds numbers, $Re_t$, for the two jets were estimated considering the thickness of the wall separating them, $h = 1$ mm, as the integral turbulent length scale, $\ell_t$ and the $u'_{inj}$ at the inlets, estimated as $u'_{inj} = 5\% \, U_{inj}$. The estimated Kolmogorov length scales for the two jets are also reported.

**Table 2.** Injection data for the two reactants in the MASCOTTE C-60 test case. The density values refer to the Peng–Robinson EoS in its enhanced translated volume formulation (the errors with respect to NIST reference data are 0.35% and 0.52%, respectively, for $O_2$ and $H_2$).

| Pressure 60 Bar | $T_{inj}$ K | $\rho_{inj}^{PR}$ kg/m³ | $\mu_{inj}$ kg/(m·s) | $\rho U \mathcal{A}$ g/s | $\mathcal{A}$ mm² | $U_{inj}$ m/s | $u'_{inj}$ m/s | $\ell_t$ mm | $Re_t$ (−) | $\eta$ μm |
|---|---|---|---|---|---|---|---|---|---|---|
| $O_2$ | 83 | 1182.97 | $2.46 \times 10^{-4}$ | 105 | 10.1736 | 8.72 | 0.436 | 1 | 2097 | 3.2 |
| $H_2$ | 275 | 5.125 | $8.59 \times 10^{-6}$ | 42 | 53.8824 | 152.09 | 7.6 | 1 | 4534 | 1.8 |

Due to the transcritical injection of oxygen, huge property changes are expected for its stream, as shown in Figure 7, where the translated volume Peng–Robinson-based data of oxygen and hydrogen are compared with their NIST counterparts as a function of temperature at 60 bar. The wide range of the compressibility factor experienced in the present simulation, [0.18–1.02] as shown in Figure 8, makes clear that the adoption of a real gas equation of state is mandatory. In the liquid oxygen jet core, it varies in the range [0.23 : 0.25] due to some pressure fluctuations; moving towards the liquid interface, there is a very thin thermal layer where the oxygen is heated and this makes Z decrease from 0.23 down to 0.18 before quickly increasing again up to 1; slightly higher values, 1.02, are reached in the hydrogen jet.

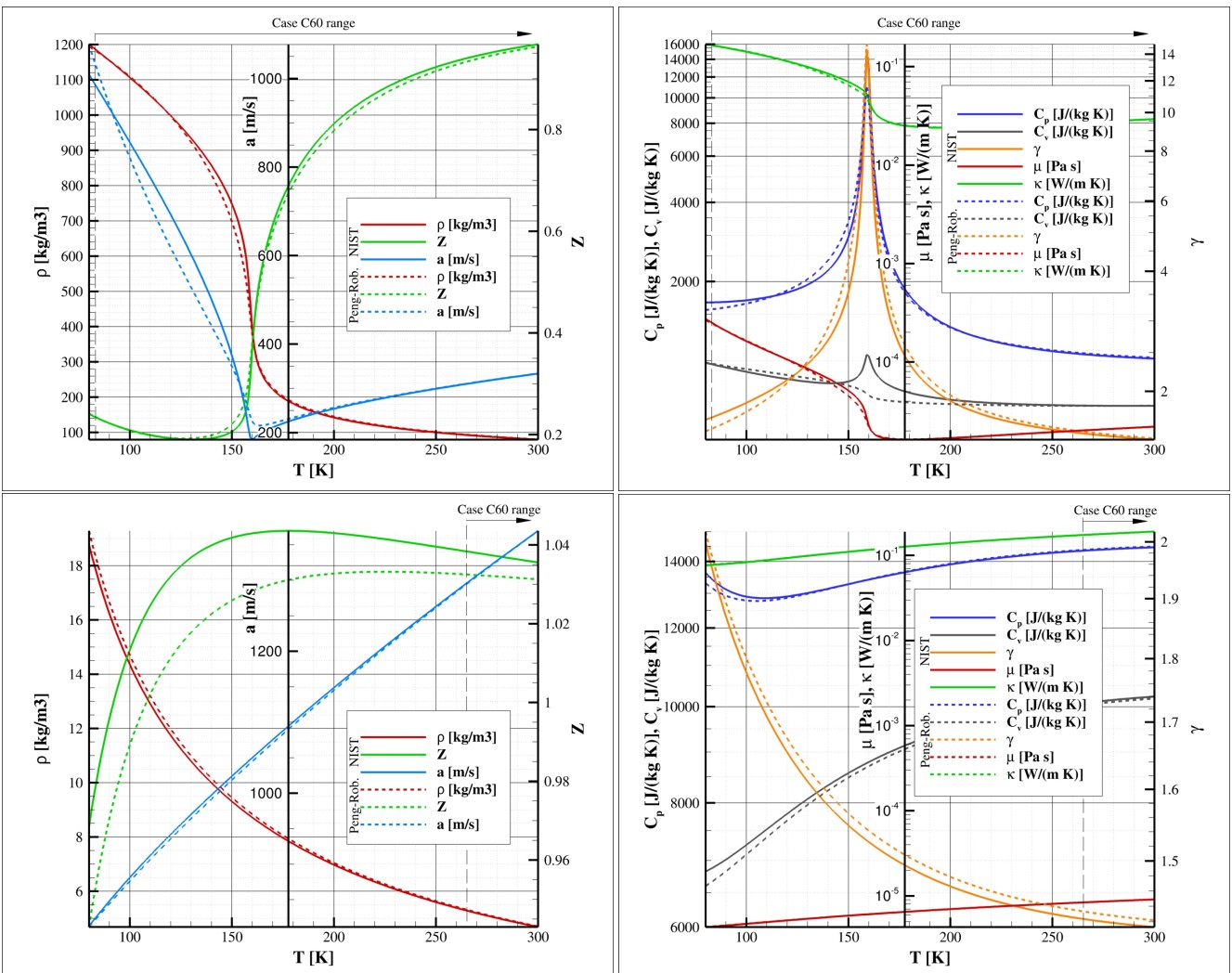

**Figure 7.** Comparison of some real gas properties estimated through the translated volume Peng–Robinson EoS and NIST reference data for $O_2$ (**top**) and $H_2$ (**bottom**) at 60 bar as in the MASCOTTE C-60 test case: density $\rho$, sound speed a, compressibility factor Z, specific heat ratio $\gamma = C_p/C_v$, viscosity $\mu$, thermal conductivity $\kappa$.

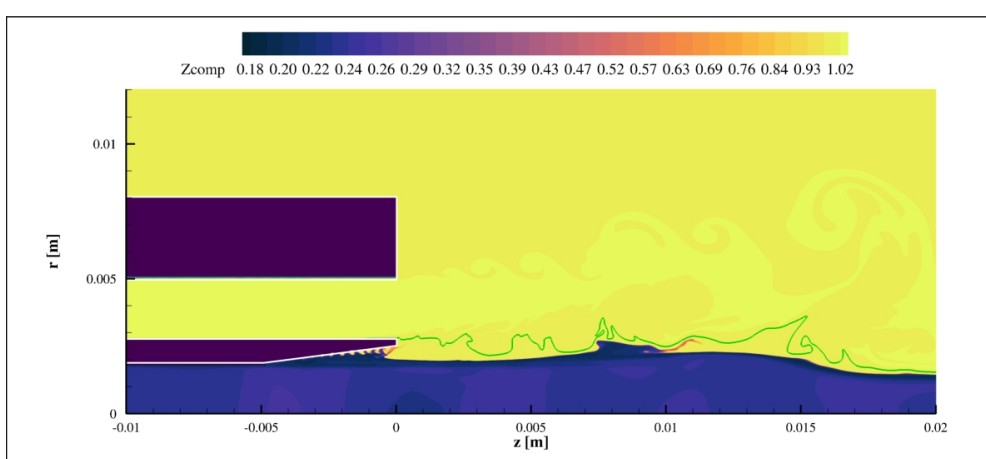

**Figure 8.** Compressibility factor $Z$ for the instantaneous field of Figure 9. The stoichiometric mixture fraction isoline $\mathcal{Z} = 0.11$ is also shown in green.

## 5. Numerical Set-Up for the 3D Les

In the simulation, the combustor is modeled as cylindrical (instead of its squared section) and the tapered central pipe is modeled in a staircase fashion, as shown in Figure 6 (right). The cooling helium film injection along the combustor walls adopted in the experiments is neglected. The computational domain simulated in this work is a 120° slice of the full cylindrical geometry. The simulated length of the combustor is 0.1 m (shorter than the experimental combustor and without its final nozzle) and that of the injector coaxial pipes is 0.01 m, as sketched in Figure 6. The domain is discretised by means of 18,425,760 grid nodes using 32 nodes along the azimuthal direction: 640,480 nodes for the $O_2$ pipe (210 nodes in the streamwise direction, $z$); 799,680 nodes for the $H_2$ pipe ($210 \times 119$ being the $z \times r$ section); 16,135,680 nodes for the combustion chamber, $210 \times 136$ being the $z \times r$ section of the zone beside the injector and $991 \times 480$ being the $z \times r$ section of the zone facing the injector. The grid has $\Delta z \in [20 : 83]$ μm up to $z = 0.02$ m, increasing up to 540 μm at $z = 0.1$ m and $\Delta r \in [43 : 60]$ μm up to $r = 0.008$ m (outer diameter of the $H_2$ injector), increasing up to 224 μm at the combustor wall.

The combustion of hydrogen and oxygen is modeled by means of the finite-rate detailed chemical mechanism derived by Boivin from the San Diego mechanism, further simplified by taking the high-pressure limit of the falloff reactions (due to the high pressure of the test case) [63,64]. The mechanism accounts for eight species, listed in Table 1 (excluding $N_2$) and 12 elementary reactions. Above 2500 K, chemical kinetics considerably affects combustion dynamics, since the $H_2 - H_2O$ equilibrium shifts towards $H_2$ and the radical pool has a significant impact in limiting the heat release. Therefore, it is necessary to adopt mechanisms including dissociation reactions and radical species, like the chosen one.

The computational domain related to the coaxial pipes was filled in with the associated incoming reactant, i.e., $H_2$ or $O_2$, while the combustor in the simulation was initially filled in with a mixture of hot products coming from a stoichiometric $H_2/O_2$ premixed flame calculation at 60 bar. Starting the simulation from this initial field, the cold reactants entered into the combustor and the mixing process was simulated in a two-dimensional axi-symmetric framework. Then, ignition was achieved by locating a spark in the mixing layer between the two reactants at ∼1.3 mm from the tip of the injector. The diffusion flame was developed in a 2D framework later copied and rotated to generate the initial three-dimensional field for the final LES simulation.

Boundary conditions at the inlets of the two jets are reported in Table 2. A synthetic turbulence generator [58] is adopted at the hydrogen flow inlet, forcing a turbulent velocity fluctuation $u'_{inj} = 5\% \, U_{inj}$ and spatial correlation length scales $\ell_z = 0.40$ mm, $\ell_r$ varying from 0.20 mm at the lower wall up to 0.25 mm at the upper wall, $\ell_\theta = 0.25$ mm, respectively, in the streamwise, radial and azimuthal directions. Without inlet turbulence, the $H_2$ jet exhibits a long turbulence transition with a nearly zero spreading angle; including

inlet turbulence, it exhibits vortex shedding, a quick turbulence transition and a non-zero spreading angle, as expected from experiments.

While the combustor walls can be assumed adiabatic, the choice of the most suitable boundary condition for the injector walls requires some reckoning based on thermal effusivity. This quantity is a measure of the ability of a material (fluid or solid) to exchange thermal energy with its surroundings and it is defined as $e = (\rho C_p \kappa)^{1/2}$, $\rho$ being the density, $C_p$ the specific heat and $\kappa$ the thermal conductivity of the material. The higher the effusivity, the higher the ability of the material to thermally influence its surroundings. Introducing the fluid and wall effusivities, it is possible to define the fluid effusivity ratio $\kappa_f = e_w/e_f$. If $\kappa_f \gg 1$, temperature at the fluid/solid interface is mostly determined by the wall temperature (that has to be evaluated) and this can be considered as isothermal; if $\kappa_f \ll 1$, the interface temperature is controlled by the fluid and the wall may be considered adiabatic; intermediate values of $\kappa_f$ correspond to walls that are neither adiabatic nor isothermal. Looking at the data in Table 3, it is observed that $\kappa_f \gg 1$ in every condition met in the test case. Hot gases in the table refer to the hot products facing the tip of the $O_2$ injector.

Upon this reckoning, the adiabatic wall boundary condition was not applied for the coaxial injector; a more accurate solution was achieved by solving the heat transfer in its solid walls. Temperature distribution in the solid was initially calculated assuming an instantaneous flowfield with a flame well anchored at the tip of the oxygen injector: the flowfield was kept constant while temperature in the solid was evolved in time (with a very high CFL, 20,000, allowing large time steps). At ~15 s from ignition, the average temperature of the tip region of the $O_2$ injector reaches ~1500 K, explaining the maximum duration of the experiment (<15 s) reported in [62]. This temperature distribution in the solid zones was assumed as initial distribution for the three-dimensional LES simulation.

**Table 3.** Estimation of effusivity fluid ratio for the fluids adjacent to the tip of the tapered lip oxygen injector. Effusivities are in SI units. Characteristic times are also reported and compared.

| Fluid/Wall | $\rho$ (kg/m$^3$) | $C_p$ (J/(kg K)) | $\kappa$ (W/(m K)) | $e_{f,w}$ (SI) | $\kappa_f$ (−) | $\tau^*$ (s) |
|---|---|---|---|---|---|---|
| H$_2$ | 5.125 | 14,414.31 | 0.1786 | 114.86 | 353.85 | $\tau_{conv} \sim 3.3 \times 10^{-5}$ |
| O$_2$ | 1182.97 | 1566.48 | 0.1636 | 550.61 | 78.81 | $\tau_{conv} \sim 5.7 \times 10^{-4}$ |
| Hot Gases | 4.55 | 2393.21 | 0.3262 | 60 | 677.38 | $\tau_{T,lip} \sim 3.8 \times 10^{-4}$ |
| Steel AISI4000 | 7850 | 4750 | 44.30 | 40,642.79 | - | $\tau_{w,lip} \sim 7.6 \times 10^{-2}$ |

Table 3 also reports the characteristic times involved in the heat transfer at the walls of the coaxial injector: the heat diffusion time in the solid wall is estimated as $\tau_{w,lip} = \delta_{lip}^2/\alpha_w$, $\delta_{lip}$ being the thickness of the oxygen injector at its tip (0.3 mm), $\alpha_w = \kappa/(\rho C_p)$ the thermal diffusivity of the material; $\tau_{conv} = L_{inj}^{div}/U_{inj}$ are the convective times of the hydrogen and oxygen streams, $L_{inj}^{div} = 4.98$ mm being the length of the divergent part of the injector; $\tau_{T,lip}$ is the lowest characteristic time of temperature dynamics in the region in front of and very close to the tip of the oxygen injector. The latter time was estimated by means of the FFTs of temperature at two locations, $m_1$ and $m_2$ on the central azimuthal plane of the simulation, having $(z; r)$ coordinates $(0.07; 2.48)$ and $(0.07; 2.85)$ (in mm), respectively, and taking the average of the inverse of the two lowest frequencies. Since the heat diffusion time in the solid wall is much longer than the fluid times, it can be concluded that the heat transfer dynamics into the solid wall is fully decoupled from the fluid dynamics.

## 6. Flame Anchoring Dynamics from the 3D LES

In this section, the characteristics of the flow and flame topology will be explored. The instantaneous fields of temperature, oxygen and hydrogen mass fractions obtained from the present simulations and reported in Figures 9, 14 and 17c can be assumed as representative of the reacting flow investigated in this section. The central liquid oxygen jet slowly enters

into the combustion chamber through its tapered injector where it exhibits separation in the divergent part. The coaxial gaseous hydrogen jet enters faster, thus producing a reacting inner shear-layer (on the oxygen side) and a non-reacting outer shear-layer. Macroscopic recirculations in the chamber bring hot products back towards the injection on the outer shear-layer side.

Since at supercritical conditions no atomisation occurs, the flame evolves in a very thin mixing layer between the gaseous hydrogen and the liquid oxygen jets. As a consequence, even for turbulent cases the flame thickness remains limited, and the flame front is confined close to the LOX jet; hence, due to the fluctuation of this thin reacting layer, it is expected that experimentalists measure low mean temperatures although instantaneous fields can exhibit much higher peaks. At the opposite, as observed in [62], a better LOX atomisation efficiency is achieved at subcritical pressures, generating lots of ligaments and droplets at the LOX jet periphery and leading to a different heat release pattern between the two regimes. CARS measurements (using $H_2$ as probe molecule) reported in [62] for a MASCOTTE test case, similar to the present one, confirmed this trend and showed a more stratified flow with much lower temperature at supercritical pressures over 50 mm downstream from the injector; at larger axial distance, the mixing between LOX and $GH_2$ is improved and the flame front becomes thicker. The present test case exhibits a similar evolution.

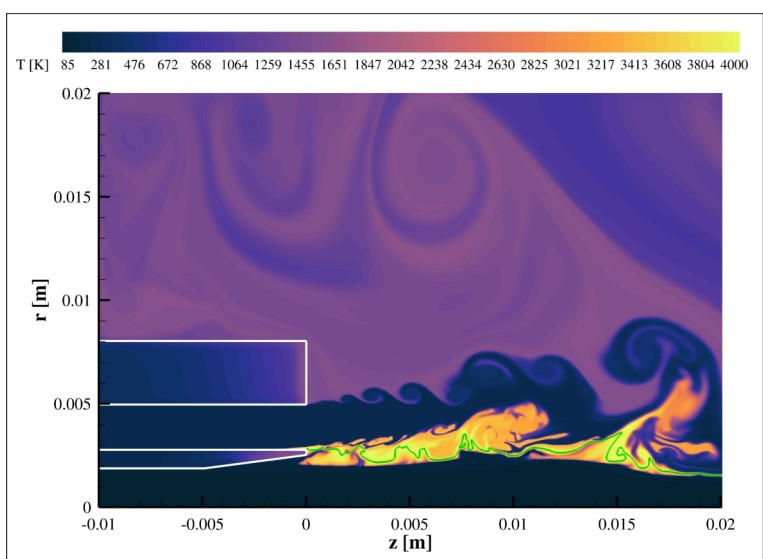

**Figure 9.** Numerical instantaneous distribution of temperature. The stoichiometric mixture fraction isoline $\mathcal{Z} = 0.11$ is also shown in green.

Figure 9 reports a snapshot of the temperature field, showing that the flame develops along the stoichiometric mixture fraction ($\mathcal{Z}_{st} = 0.11$, corresponding to the equivalence ratio $\Phi = 1$) in the inner shear-layer, close to the maximum density gradient iso-surface but at greater radial distance. The flame is anchored at the outer edge of the $O_2$ injector, i.e., the inner edge of the $H_2$ injector. The vortex shedding of the fast hydrogen flow from this inner edge produces local pressure fluctuations that suck oxygen from the low speed recirculation zone of the tapered central injector up to the inner hydrogen edge (see Figure 10). Although at some phases the hydrogen stream tends to push back the sucked oxygen, this happens so mildly that a nonnegligible $O_2$ concentration is on average located around the inner edge of the $H_2$ injector. Hence, the stoichiometric mixture fraction isoline, and consequently the flame, is stably anchored at the inner edge of the $H_2$ injector. The described fluidynamic process has a short characteristic time, $\sim$23.12 μs, corresponding to $\sim$43,250 Hz.

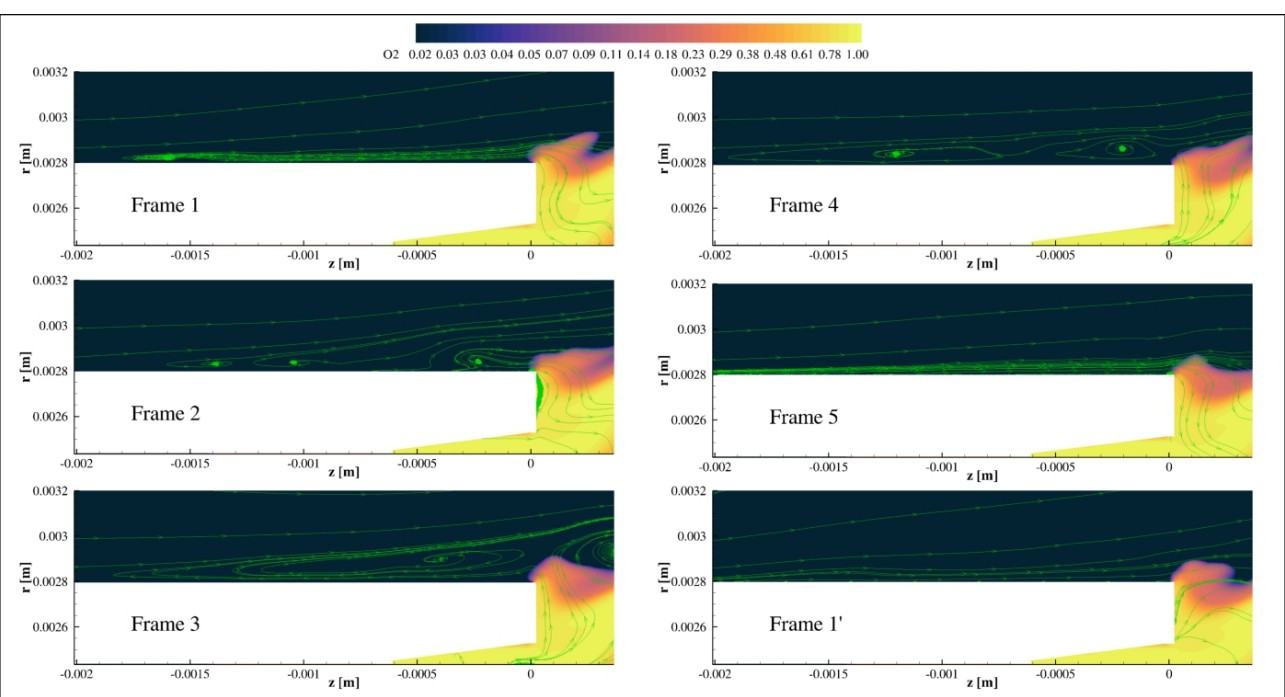

**Figure 10.** Typical evolution of streamlines and oxygen distribution at the tip of the oxygen injector, showing how the $O_2$ is brought to the inner edge of the hydrogen injector. The delay time between frames 1 and 5 is $\sim$23.12 μs, corresponding to $\sim$43,250 Hz.

Since the $H_2$ flow tends to separate along its inner wall over nearly 2 mm from the exit, the flame goes upstream along this wall for such a length, thus providing a continuous ignition of the flame. The interaction between the shedding of large structures at the outer edge of the $H_2$ pipe and the synchronous shedding of small vortices at its inner edge produces a complex dynamics of the flame. The vortices shed at high frequency from the $H_2$ inner wall produce intense mixing of fresh reactants with hot products, thus promoting combustion; at the same time, their flame stretching also produce localised extinctions, resulting in flame tongues released inside the hydrogen jet. Due to the synergic action of vortex shedding on the $H_2$ side and the vortices in the wake of the separated region downstream of the $O_2$ injector divergent part, the flame is not a very thin layer; the reacting region has a thickness comparable to the injector wall separating the $O_2$ and $H_2$ streams. While rapidly expanding, largely wrinkled by coherent structures and turbulence with wavelengths increasing with the axial distance from injection, the flame pushes the $H_2$ jet away from the axis. Figure 11 reports the temperature evolution at three instants having a delay of $\sim$11 μs, evidencing what was previously described.

Figure 12 shows a zoomed-in view of the equivalence ratio distribution close to the injector. It is observed that most of the combustion takes place at rich fuel conditions (equivalence ratio $\Phi > 1$) and where the compressibility factor is 1, i.e., the gas can be assumed ideal, at least from the point of view of the equation of state (the same cannot be said for the real gas transport properties which typically differ from the ideal ones).

A quantitative description of the non-premixed flame structure is provided in Figure 13, showing an instantaneous scatter plot of the OH and $H_2O_2$ radical species versus the mixture fraction and parameterised with temperature. Peaks of both radicals are around the stoichiometric mixture fraction, with the OH peak being slightly closer to the lean side. The OH radical is an important species for the ignition delay time. The radical $H_2O_2$, commonly known as "radical scavenger", terminates activated radicals like OH, H, O: this happens close

to the oxygen jet and in the fuel rich regions due to their low temperatures. Chemical reactions involved in the adopted Boivin's mechanism [63,64] and responsible for the termination are:

$$2\,HO_2 \rightarrow H_2O_2 + O_2$$
$$HO_2 + H_2 \rightarrow H_2O_2 + H. \tag{14}$$

The first reaction has a low activation energy ($\sim$1385 cal/mol), while the dissociation reaction of $H_2O_2$, i.e.,

$$H_2O_2 + M \rightarrow 2\,OH + M, \tag{15}$$

has the highest activation energy ($\sim$51,300 cal/mol) among the reactions involved in the mechanism. Hence, $H_2O_2$ removes the $HO_2$ radical mainly due to the first of the chemical reactions (14). Consequently, the OH radical formation through the chemical reaction

$$HO_2 + H \rightarrow 2\,OH \tag{16}$$

is not promoted, although it has very low activation energy ($\sim$294 cal/mol). Once the $H_2O_2$ is formed, it diffuses into the $H_2$ (mixture fraction > 0.6 ) and $O_2$ streams (mixture fraction < 0.025), as shown in the scatter plots.

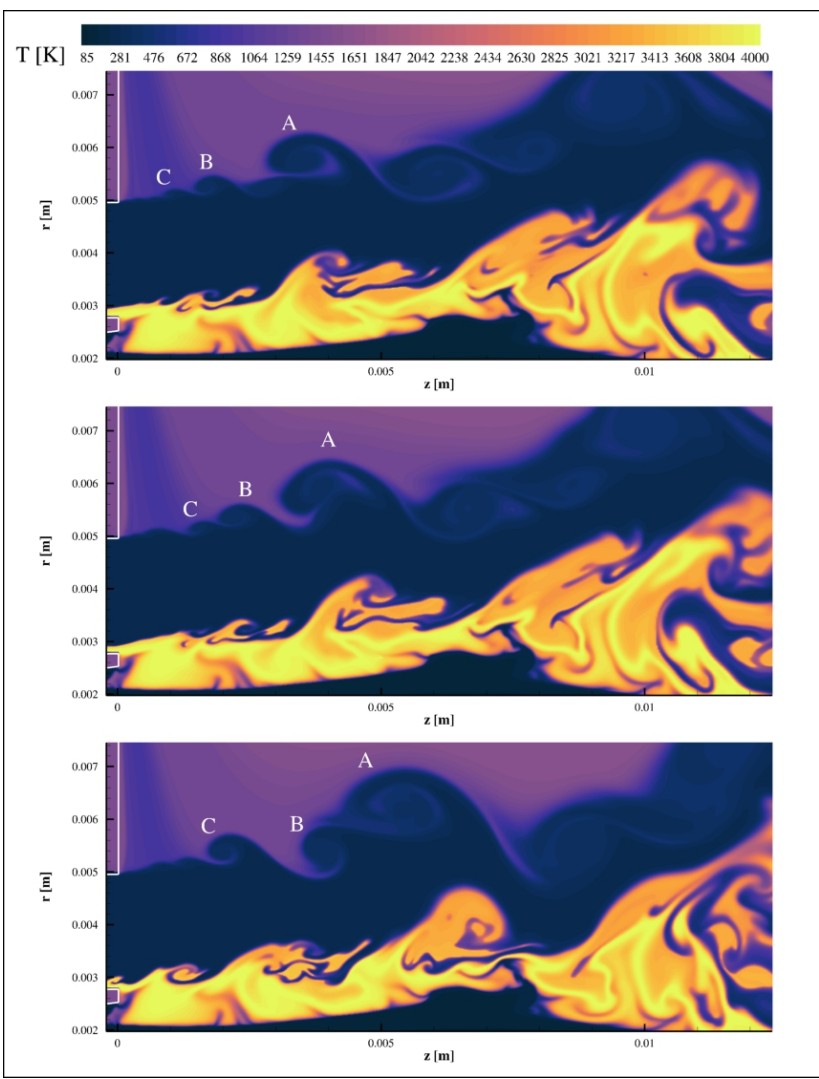

**Figure 11.** Instantaneous temperature fields close to the injection, evidencing the inner and outer shear-layers; three coherent structures are marked with the labels A, B, C. The delay time between each frame (from top to bottom) is $\sim$5.74 µs.

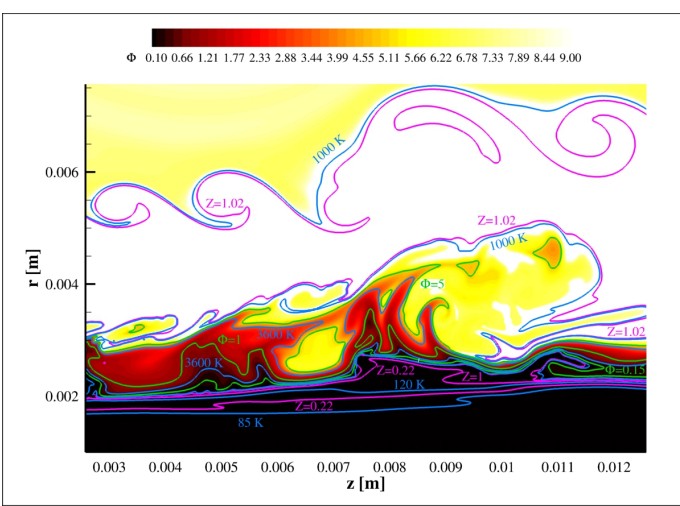

**Figure 12.** Instantaneous zoomed-in view of the equivalence ratio $\Phi$ close to the injector: three levels, 0.15, 1 (corresponding to the stoichiometric mixture fraction $\mathcal{Z} = 0.11$) and 5 are marked as green iso-lines. Some temperature iso-lines are shown in blue, while some compressibility factor iso-lines are in purple.

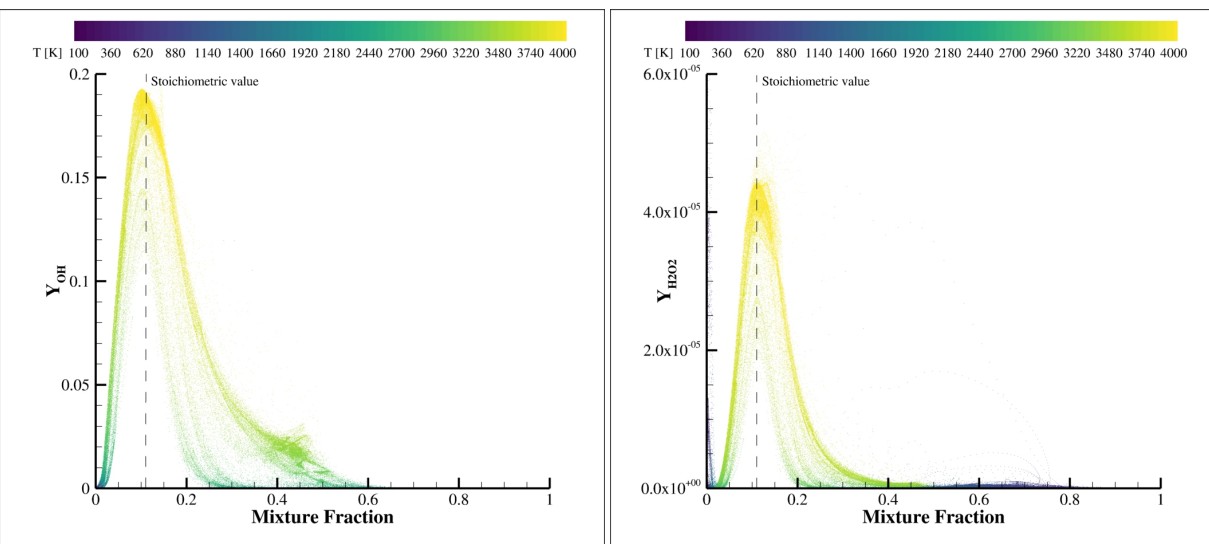

**Figure 13.** Instantaneous scatter plots of OH and $H_2O_2$ mass fractions versus the mixture fraction and parameterised with temperature; the stoichiometric mixture fraction $\mathcal{Z} = 0.11$ is marked as a dashed line.

### 6.1. The Gaseous Hydrogen Jet

The hydrogen stream is faster than oxygen; a snapshot is shown in Figure 14. The jet exhibits high-frequency vortex shedding ($\sim$87 kHz, corresponding to a Strohual number $St\sim$1.1) from both the inner and outer edges of its injector, as shown by the vorticity module dynamics in Figure 15. It is observed that without inlet turbulence, the $H_2$ jet exhibits a long turbulence transition with a nearly zero spreading angle.

The coherent structures released in the outer shear-layer are clearly visible; they are released from the outer edge of the $H_2$ injector; they grow moving downstream and quickly coalesce, evolving to developed turbulence and feeding the recirculation regions of the combustor generated by the impingement of the hydrogen jet on the chamber walls. Such recirculations bring hot products back to the inlet side. The structures released in the inner shear-layer are less coherent and smaller, with respect to the size of the tip of the $O_2$ injector; they are released before the inner edge of the $H_2$ injector due to the separation of the flow evidenced in Figure 10; they grow less and live shorter, being damped by the flame.

It is observed that the velocity fluctuations in the $H_2$ jet region in the combustion chamber are largely enhanced with respect to the inlet turbulence level of 5%, reaching levels above 30%, as shown in Figure 16.

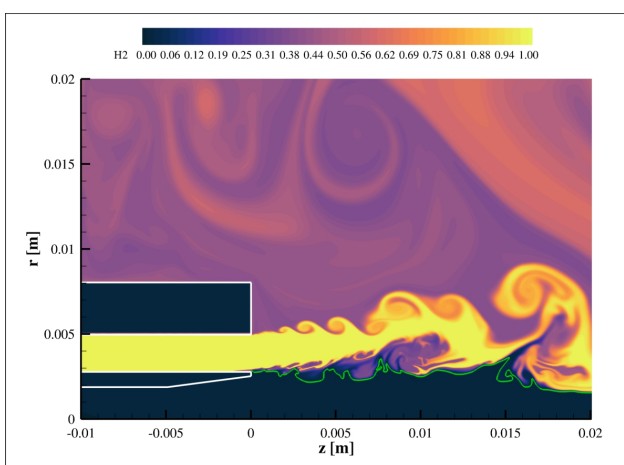

**Figure 14.** Numerical instantaneous distribution of the $H_2$ mass fraction at the same time as Figure 9; the stoichiometric mixture fraction isoline $\mathcal{Z} = 0.11$ is also shown in green.

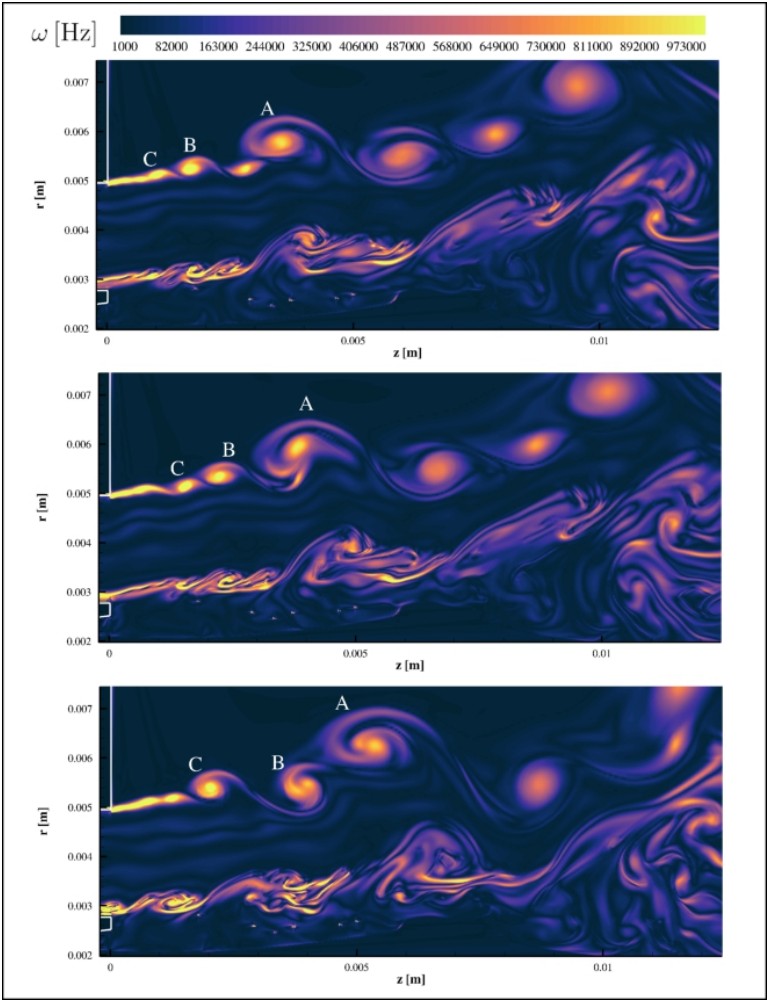

**Figure 15.** Instantaneous fields of vorticity module at the same times as Figure 11 close to the injection; three coherent structures are marked with the labels A, B, C. The delay time between each frame (from top to bottom) is $\sim$5.74 $\mu$s. The three frames nearly cover a vortex shedding period ($\sim$11.48 $\mu$s).

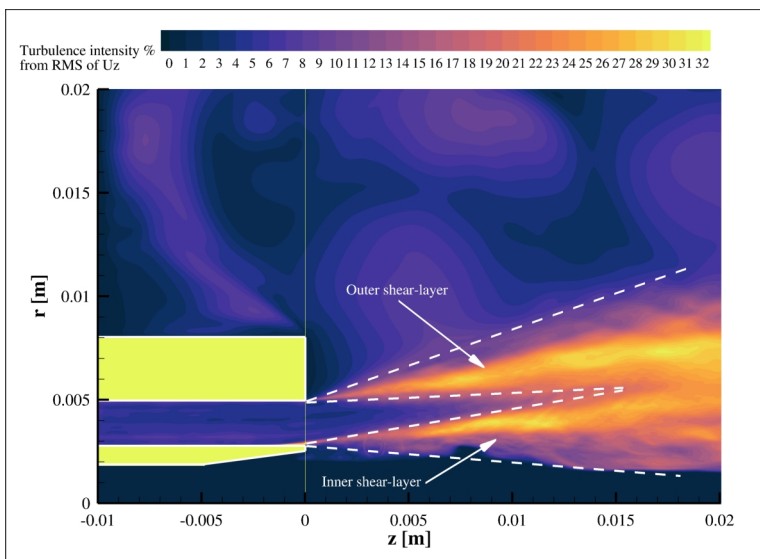

**Figure 16.** Turbulence intensity percentage of streamwise velocity, $U_z$, calculated from its RMS fluctuations. Both inner and outer shear-layers are clearly identified.

### 6.2. The Liquid Oxygen Jet

For a supercritical pressure, LOX is known to be a dense medium with physical properties qualitatively similar to that of a gaseous medium. Authors in [62,65] found only small wrinkles at the LOX jet surface, as shown in Figure 17a,b, similar to those generated by conventional turbulent mixing layers between two compressible gases; they did not find any ligaments or droplets at the periphery of the cryogenic jets they examined, at least with the attainable spatial resolution (40 micron in a first study, 20 micron later), in agreement with previous findings [6,66]. The present simulation has a spatial resolution similar to the experimental work and is in agreement with its findings: the high-density liquid $O_2$ jet largely and slowly penetrates the combustion chamber along its axis, showing a weakly turbulent interface with some structures protruding or diffusing into the surrounding $H_2$ jet and reacting region (see Figure 17c). As in Mayer's test case 3, the $O_2$ jet penetration is favoured by the $C_p$ surge across the pseudo-boiling line (see Figure 7 top-right) also in this case.

It is observed that the liquid oxygen interface appears more wrinkled and with more sharply defined structures in the non-reacting condition, i.e., when no ignition is provided to reactants: in this case, the inner shear-layer is characterised by high-momentum ligaments protruding from the liquid oxygen interface into the surrounding hydrogen jet, as shown in Figure 18, thus enhancing turbulent mixing. In a nutshell, in the non-reacting case the two streams of reactants form a high velocity gradient shear-layer promoting mixing and shortening the penetration depth of the liquid $O_2$ jet, while in the reacting case the expansion caused by pseudo-boiling and combustion heat release push the $H_2$ jet away from the centerline, largely reducing the turbulent mixing. These deductions are in agreement with what was reported in [67].

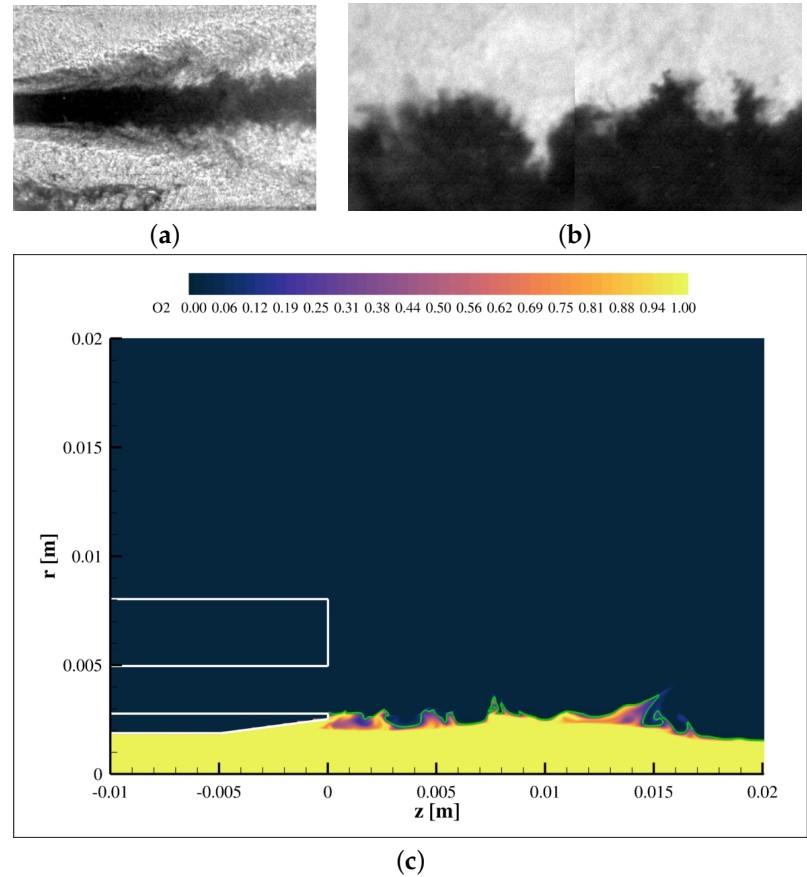

**Figure 17.** At the top, shadowgraphy (**a**) and back-lighting (**b**) experimental images showing a typical instantaneous structure of the liquid $O_2$ jet for the present investigated case [62]. At the bottom, numerical instantaneous distribution of the $O_2$ mass fraction (**c**) at the same time as Figure 9; the stoichiometric mixture fraction isoline $\mathcal{Z} = 0.11$ is also shown in green.

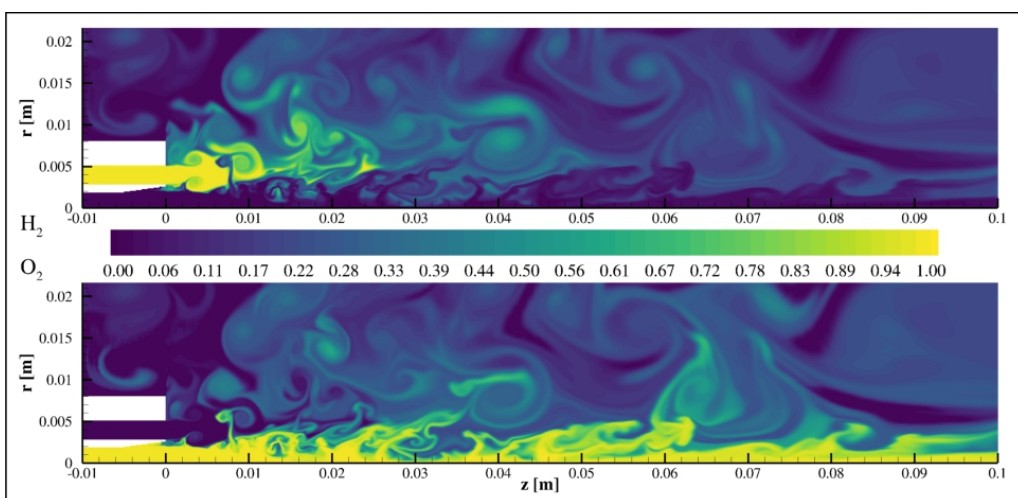

**Figure 18.** Numerical instantaneous distribution of hydrogen and oxygen mass fractions for the investigated flow without ignition of reactants, i.e., with a non-reacting shear-layer between the liquid $O_2$ and the gaseous $H_2$ jets.

Based on what was said in the introduction, solving the liquid oxygen interface without numerical oscillations requires robust and accurate numerical schemes. To provide an example of the difficulties experienced in such simulations, Figure 19 shows a zoomed-in

view of the equivalence ratio distribution close to the injector, more zoomed in than in Figure 12. It is observed that the liquid oxygen is preheated by the surrounding fluid in a very thin layer, causing a sharp change in its compressibility factor, decreasing from 0.22 to 0.18 and then increasing very quickly to 1. The same figure also shows the presence of strong temperature gradients.

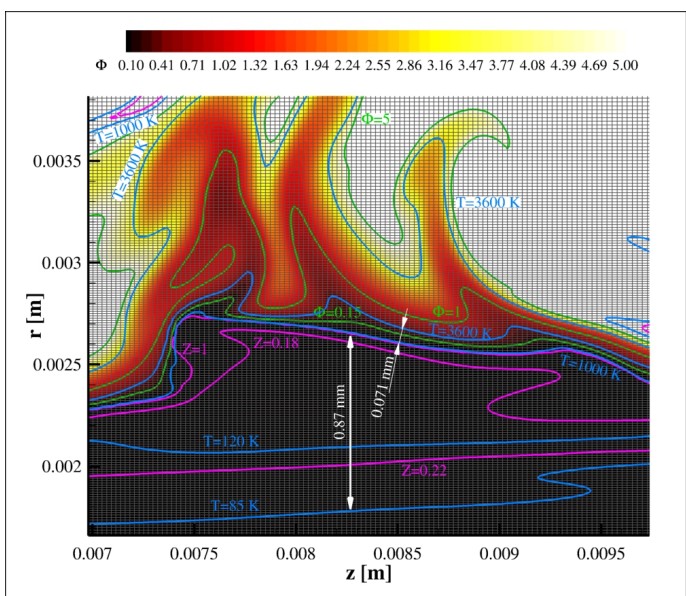

**Figure 19.** Instantaneous zoomed-in view of the equivalence ratio $\Phi$ close to the injector: three levels, 0.15, 1 (corresponding to the stoichiometric mixture fraction $\mathcal{Z} = 0.11$) and 5 are marked as green iso-lines. Some temperature iso-level are shown in blue, while some compressibility factor iso-lines are in purple.

## 7. Conclusions

The article has initially validated the numerics and real gas models implemented in the HeaRT code adopted in this work. Such validation was achieved by simulating a non-reactive test case focused on the injection of a liquid nitrogen jet into a chamber filled in with gaseous nitrogen at ambient temperature. The obtained numerical data compared well with experimental ones, evidencing the robustness and the goodness of the numerics and physical models implemented.

After this validation, a MASCOTTE test case dealing with combustion of liquid oxygen and gaseous hydrogen at 60 bar was simulated, focusing on the flame anchoring mechanism and the characteristics of the two reactant jets.

Heat transfer into the solid walls of the coaxial injector was solved to provide a more realistic boundary condition and a more reliable investigation on the flame anchoring. The flow exhibits a non-reacting outer shear-layer and a reacting inner shear-layer, the latter close to the liquid oxygen interface. The outer shear-layer is characterised by large scale coherent turbulent structures released by the outer edge of the hydrogen injector. The high-frequency vortex shedding from the inner edge of the hydrogen injector produces a quite stable oxygen rich region close to this edge. Hence, the flame is attached at this edge and, due to the separation of the hydrogen stream close to the end of its inner wall, it propagates back along that wall. The non-premixed flame is largely stretched by the released eddies in the inner shear-layer, where reacting tongues consequently appear. Most of combustion is fuel rich.

The liquid oxygen interface was analysed, evidencing a weakly turbulent surface with some structures protruding or diffusing into the surrounding hydrogen jet and reacting region. Liquid oxygen is preheated in a very thin layer, exhibiting a sharp change in the compressibility factor and temperature. Solving such strong spatial gradients proved the robustness of the numerical approach.

**Author Contributions:** Methodology, investigation, writing, E.G., D.C. and N.A.; software, all. All authors have read and agreed to the published version of the manuscript.

**Funding:** This research received no external funding.

**Data Availability Statement:** Data can be asked to the authors.

**Acknowledgments:** The computing resources and the related technical support used for this work have been provided by CRESCO/ENEAGRID High Performance Computing infrastructure and its staff [45]. CRESCO/ENEAGRID High Performance Computing infrastructure is funded by ENEA, the Italian National Agency for New Technologies, Energy and Sustainable Economic Development and by Italian and European research programmes (see http://www.cresco.enea.it/english (accessed on 8 October 2022)).

**Conflicts of Interest:** The authors declare no conflict of interest.

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
