# Peer review of "Flame Anchoring of an H2/O2 Non-Premixed Flamewith O2 Transcritical Injection"

_aerospace, doi:10.3390/aerospace9110707_

Round 1

Reviewer 1 Report

This is an interesting paper on the transcritial combustion phenomenon. I have a few concerns here:

turbulent chemistry interaction closure is an important part of the study and the authors should elaborate on alternative models found in the literature and any relative comparison of these models with that of the authors.

in section 3, it would be good to also demonstrate the validity of LES with the used Grid.

The authors explained the reactive case mostly through qualitative demonstrations. some figures like scatter plots of some important species along with temperature versus equivalence ration can compensate for such lack of analysis.

Author Response

To all reviewers:

The authors noted that the mixture fraction was wrongly calculated in the previous version of the article.

Hence, all the pictures involving the mixture fraction have been updated (also those having the stoichiometric line only).

REV1

1. turbulent chemistry interaction closure is an important part of the study and the authors should elaborate on alternative models found in the literature and any relative comparison of these models with that of the authors.

1. The authors agree with the reviewer about the importance of the turbulence/combustion closure adopted.

The LTSM model has been developed and validated in a previous paper. 

In literature there exist several sub grid scale (SGS) models (mostly based on eddy viscosity), both in their static and dynamic formulation, that are specific for subgrid stresses and subgrid heat fluxes.

For example, among the SGS stresses models there are the Smagorinsky model, the WALE model, the structure function models of Lesieur, the mixed scale model, structural models like deconvolution and scale-similarity models.

Models for SGS heat and mass fluxes are typically based on the subgrid Prandtl and Schmidt numbers, respectively.

The interaction between turbulence and combustion is devoted to the closure of the subgrid chemical source term. Also in this case there exist specific models, like the eddy-break up model, the eddy dissipation concept (EDC), the Linear Eddy Model (LEM), the surface density model (for high Damkoehler numbers), the thickened flame model.

A comparison of the results obtained in this paper by means of the LTSM model with those obtained by means of other turbulent combustion models would be interesting. However, the work required is not negligible; we could face it in another paper.

2. in section 3, it would be good to also demonstrate the validity of LES with the used Grid

2. The quality index suggested by Pope has been calculated for the non reacting case presented in Section 3. Such index, IQ_Pope = K_rms / (K_rms + K_sgs), is > 84% in most of the computational field; the suggested limit for a reliable LES is 80%.

Lines 204-207 were added in red in the text.

3. The authors explained the reactive case mostly through qualitative demonstrations. some figures like scatter plots of some important species along with temperature versus equivalence ration can compensate for such lack of analysis.

3. As suggested by the reviewer, the authors added two figures (Fig. 13 in the new version) showing the scatter plots of two radical species, OH and H2O2, versus the mixture fraction and coloured by temperature. These figures contribute to a quantitative description of the structure of the non-premixed flame investigated. They were commented in the text of the article 

Adding a new paragraph at the end of Section 6 (from line 369 in red).

Reviewer 2 Report

Comments

1.     Line 55-57, Authors kindly use alternative terms for “ some, someone, someothers, sometimes, some authors”, specify author names. English writing must be improved.

2.     Line 65-69, Authors kindly explain specifically what the exact missing research gap is addressed in this paper.

3.     Table 1 header, correct the typo “exclamatory sign is present”

4.     Authors can quantify the results for anchoring distance, temperature distribution and which parameter is sensitive out of your analysis, for example you can vary the inlet turbulence, equivalence ratio  and describe its effect.

5.     Line 435, correct the typo “proved”

Author Response

To all reviewers:

The authors noted that the mixture fraction was wrongly calculated in the previous version of the article.

Hence, all the pictures involving the mixture fraction have been updated (also those having the stoichiometric line only).

REV2

1.     Line 55-57, Authors kindly use alternative terms for “ some, someone, someothers, sometimes, some authors”, specify author names. English writing must be improved.

1. As suggested, the authors changed lines 55-59 ( in red).

2.     Line 65-69, Authors kindly explain specifically what the exact missing research gap is addressed in this paper.

2. As suggested by the reviewer, the authors explained the missing research gap addressed in the paper, by adding lines 68-72  in red  in the text.

3.     Table 1 header, correct the typo “exclamatory sign is present”

3. Corrected.

4.     Authors can quantify the results for anchoring distance, temperature distribution and which parameter is sensitive out of your analysis, for example you can vary the inlet turbulence, equivalence ratio  and describe its effect.

4. The effect of not including turbulence at the inlet of the hydrogen jet was investigated in the work. This was written  in red at lines 275-278. A new figure could be added but it does not add anything else apart from a visual representation of what already described.

(effect on the H2 stream evolution and spreading angle). Now this is remarked in red in Section 6.1 (lines 376-377) related to the analysis of the H2 jet.

5.     Line 435, correct the typo “proved”

5. Corrected.

Round 2

Reviewer 1 Report

the authors have met my concerns.

Reviewer 2 Report

Comments are addressed